# Cefiderocol in Critically Ill Patients with Multi-Drug Resistant Pathogens: Real-Life Data on Pharmacokinetics and Microbiological Surveillance

**DOI:** 10.3390/antibiotics10060649

**Published:** 2021-05-28

**Authors:** Christina König, Anna Both, Holger Rohde, Stefan Kluge, Otto R. Frey, Anka C. Röhr, Dominic Wichmann

**Affiliations:** 1Department of Intensive Care Medicine, University Medical Center Hamburg-Eppendorf, 20251 Hamburg, Germany; ch.koenig@uke.de (C.K.); s.kluge@uke.de (S.K.); 2Department of Pharmacy, University Medical Center Hamburg-Eppendorf, 20251 Hamburg, Germany; 3Department of Microbiology, University Medical Center Hamburg-Eppendorf, Virology and Hygiene, 20251 Hamburg, Germany; a.both@uke.de (A.B.); h.rohde@uke.de (H.R.); 4Department of Pharmacy, General Hospital of Heidenheim, 89522 Heidenheim, Germany; otto.frey@kliniken-heidenheim.de (O.R.F.); Anka.Roehr@Kliniken-Heidenheim.de (A.C.R.)

**Keywords:** cefiderocol, therapeutic drug monitoring, pharmacokinetics, critically ill patients, continuous veno-venous hemodialysis

## Abstract

Cefiderocol is a new siderophore-cephalosporin for the treatment of multi-drug resistant Gram-negative pathogens. As a reserve agent, it will and should be used primarily in critically ill patients in the upcoming years. Due to the novelty of the substance little data on the pharmacokinetics in critically ill patients with septic shock and renal failure (including continuous renal replacement therapy and cytokine adsorber therapy) is available. We performed therapeutic drug monitoring in a cohort of five patients treated with cefiderocol, to improve the knowledge on pharmacokinetics in this vulnerable patient population. As expected for a cephalosporin with predominantly renal elimination the maintenance dose could be reduced in patients with renal impairment or on continuous renal replacement therapy. The manufacturer’s dosing instructions were sufficient to achieve a drug level well above the MIC. However, the addition of a cytokine adsorber might reduce serum levels substantially, so that in this context therapeutic drug monitoring and dose adjustment are recommended.

## 1. Introduction

Cefiderocol is a novel siderophore-cephalosporin, with broad activity against multi-drug resistant (MDR) Enterobacterales and non-fermenting Gram-negative bacteria like *Pseudomonas aeruginosa*, *Acinetobacter baumannii* or *Stenotrophomonas maltophilia*. This novel antibiotic drug, showed potent in-vitro activity against over 90% of meropenem-resistant Gram-negative bacilli [1,2]. Most importantly, susceptibility was sustained in a majority of metallo-ß-lactamase-expressing bacteria, for which few treatment options are available [3]. By the addition of a catechol side-chain cefiderocol gains the ability to act as a siderophore [4]. This facilitates the drug’s active transport into the bacterial cell and protects it from efflux mechanisms [5]. Due to the novelty, there is currently limited data on the clinical use of cefiderocol against MDR Gram-negative bacteria in critically ill patients. However, the substance appears to offer great potential, especially for these patients at high risk. As a beta-lactam antibiotic cefiderocol shows time dependent bactericidal activity with a pharmacokinetics/pharmacodynamics (PK/PD) target of 75% *f*T > MIC of 2 mg/L. In plasma approximately 50% of cefiderocol is protein-bound [6]. Cefiderocol shows linear pharmacokinetics with a half-life (t_1/2_) of 2–2.7 h and a drug clearance (CL) of 4.6–6 L/h in healthy volunteers [6]. With up to 70% excreted unchanged in the urine, reduction in drug CL and t_1/2_ prolongation is correlated to the degree of renal insufficiency. Therefore, dose adaption according to glomerular filtration rates (eGFR) estimated by the Cockcroft Gault formula is currently recommended by the manufacturer. However, altered pharmacokinetics of beta-lactam antibiotics especially in the critically ill patient cohort is a well-known problem when it comes to optimal drug dosing [7]. Apart from extended volumes of distribution and dynamic changes in drug elimination the need for extracorporeal organ support such as continuous renal replacement therapies (CRRT) affect antibiotic exposure. To date, there is no real-life data on cefiderocol dosing during CRRT. Current dosing recommendations for CRRT are based on approximations due to cefiderocol’s similarity to cefepime [6]. Additionally, high inter-individual PK-variability of antibiotics is expected in critically ill patients [8,9].

Recently presented data suggest, that dosing recommendations for cefiderocol result in adequate drug levels in plasma as well as epithelial lining fluid [10]. The use of extracorporeal organ replacement therapy may have a negative impact. In this context, therapeutic drug monitoring of cefiderocol levels in critically ill patients is a reasonable tool contributing to the safety and efficacy of the drug [11]. Here, we report the results of therapeutic drug monitoring (TDM) for cefiderocol in a cohort of five patients with infections caused by multi-drug resistant Gram-negative pathogens with severe septic shock, acute renal failure (AKI) requiring CRRT (n = 3) and additional cytokine adsorber therapy (n = 1). Along with microbiological surveillance cultures were performed to detect a potential emergence of cefiderocol resistance.

## 2. Results

### 2.1. Individual Patients

Five patients were included into the study, four were men, the median age was 53 years (range 35 to 76). At initiation of cefiderocol therapy all patients were in septic shock, received invasive ventilation and vasopressor therapy. Median SOFA and SAPS scores were 11 (range 4 to 19) and 49 (range 33 to 68), respectively underlining the severity of the disease. Continuous renal replacement therapy (CRRT) performed as hemodialysis was applied in three and additional cytokine adsorber (CytoSorb^®^, Monmouth Junction, NJ, USA) therapy in one patient.

Patient #1: A 41-year-old man presented with acute chronic respiratory failure due to a deterioration of a known pulmonary hypertension, a community acquired pneumonia and acute renal failure (CKD-EPI eGFR = 10 mL/min/1.73 m^2^). The patient was transferred to the intensive care unit (ICU) for non-invasive ventilation and intravenous treprostinil therapy. His previous medical history was notable of chronic renal impairment grade III (Acute Kidney Injury Network Criteria), an idiopathic pulmonary fibrosis requiring long term oxygen supplementation and recurrent episodes of pneumonia due to *Pseudomonas aeruginosa*. Due to septic shock, respiratory and kidney failure, invasive ventilation and CRRT were necessary on day 33. Microbiological samples revealed a MDR *Pseudomonas aeruginosa* from an urine culture and a third and fourth generation cephalosporin-resistant *Pseudomonas aeruginosa* from a respiratory culture. Therapy with cefiderocol accompanied by TDM was initiated with a dose of 2 g eight hourly administered in prolonged infusion mode according to the manufactures’ instruction. The patient improved over the next 4 days and vasopressor support as well as invasive ventilation were stopped. C-reactive protein (CRP) dropped from 293 to 16 mg/dL and the patient was transferred to the peripheral ward with normal leucocyte counts (6 Mrd/L). Since microbiological cure was achieved by the end of therapy (7 days) no sample for cefiderocol susceptibility testing was available.

Patient #2: A 69-year-old woman was admitted to the ICU for post-operative monitoring after esophagectomy for esophageal cancer. The post-operative course was complicated by insufficiency of the gastro-esophageal anastomosis, mediastinal abscess, invasive pulmonary aspergillosis and multiple episodes of septic shock due to nosocomial pneumonia and recurrent need of CRRT. Recurrent respiratory samples revealed *Pseudomonas aeruginosa* producing extended-spectrum beta-lactamases (ESBL) susceptible to meropenem and an MDR *Pseudomonas aeruginosa*. Therapy with cefiderocol adapted for renal insufficiency accompanied by TDM was initiated with a dose of 1 g eight hourly administered in prolonged infusion mode. Inflammation markers dropped (CRP 243 to 117 mg/dL) and the clinical status improved slightly over the next days with ceficerocol being stopped on day 14. Clinical improvement was achieved partially with one isolate persisting in relevant quantities in the lungs during the course of the treatment whilst maintaining susceptibility to cefiderocol. During the further course, her clinical condition did not improve substantially. Due to an unfavorable prognosis, the patient was set on comfortable care and died a few days later.

Patient #3: A 76-year-old man suffering from COVID-19 repatriated from South-Eastern Europe for extracorporeal membrane oxygenation (ECMO) for acute respiratory distress syndrome (ARDS) therapy. His medical history consisted of coronary heart diseases for which he received medical therapy. Microbiological screening on admission revealed a MDR *Acinetobacter baumannii* in a blood culture and in respiratory material. Therapy with cefiderocol accompanied by TDM was initiated with a dose of 2 g eight hourly administered in prolonged infusion mode. On day 4 CRRT was initiated due to acute renal failure. Subsequent blood cultures after treatment initiation remained negative, demonstrating microbiological cure. Despite intensive treatment, the patient died on day 9 due to progressive respiratory failure.

Patient #4: A 53-year-old man with a history of allogeneic stem cell transplantation for acute myeloic leukemia was transferred to the ICU for the treatment of a blood stream infection with a carbapenemase (Verona-Integron metallobetaclactamase = VIM) producing *Pseudomonas aeruginosa*. Due to intensive immunosuppressive therapy (ruxolitinib, methylprednisolon, mycophenolat-mofetil) for cutaneous and intestinal graft-versus-host diseases grade 4 his concomitant diagnoses included a Human-Herpes-Virus-6 encephalitis, invasive pulmonary mold infection (*Lichenhaimia* spp. and *Aspergillus fumigatus*) and a BK-virus cystitis. Therapy with cefiderocol accompanied by TDM was initiated with a dose of 2 g eight hourly according to the manufactures’ instruction. Due to metabolic deterioration CRRT was initiated on day 3. CRP levels were consistently high (254 to 367 mg/dL) and despite treatment with granulocyte-colony stimulating factor the patient remained aplastic. Whilst maintaining sufficient cefiderocol levels (C_min_ > 40 mg/L) blood cultures obtained on day 6 revealed a cefiderocol-resistant *Pseudomonas aeruginosa* (MIC of 16 mg/L). The primarily targeted isolate could not be re-isolated later in the course. Therapy was changed to colistin and meropenem but the patient died on day 8 due to progressive septic shock.

Patient #5: A 50-year-old man without preexisting medical conditions with ARDS and acute renal failure due to COVID-19 was transferred to our hospital. Treatment consisted of CRRT and extracorporeal-membrane-oxygenation (ECMO). He developed a hospital acquired pneumonia with a carbapenem-resistant *Acinetobacter baumannii* cultured in the bronchoalveolar lavage (BAL). Therapy with cefiderocol accompanied by TDM was initiated with a dose of 2 g eight hourly administered in prolonged infusion mode. After the first dose the treatment was extended by cytokine adsorber therapy with (CytoSorb^®^, Monmouth Junction, NJ, USA). The patient died on day 6 due to progressive respiratory failure whilst the *Acinetobacter baumanii* could still be isolated from BAL samples.

A summary on preexisting medical conditions, CRRT dose and the primary source of the infections is given in Table 1.

### 2.2. Microbiological Surveillance

Targeted pathogens were either *Pseudomonas aeruginosa* (n = 3) or *Acinetobacter baumannii* (n = 2) with cefiderocol MICs ranging from 0.125 to 0.5 mg/L. Thus, based on previous studies and the current EUCAST criteria (EUCAST 2021, v11), all pathogens were classified as susceptible to cefiderocol [12,13,14].

Clinical cure (a primary outcome parameter of the European Medicines Agency for approval of new antibiotics to the EU market) could be achieved in patient #1 and #2. Microbiological cure (a primary outcome parameter of the Food and Drug Agency for approval of new antibiotics in the US-market) could be achieved in patient #1, #3 and #4 with the exception that the resistant *Pseudomonas aeruginosa* in patient #4 was not the isolate targeted by the initial therapy. Patient #5 was suffering from COVID-19 and a hospital acquired pneumonia caused by *Acinetobacter baumannii*. Even though the MIC was low (0.25 mg/L), the isolate could be cultured throughout the clinical course. A graphic summary of the individual clinical courses (inflammatory markers, vasopressor- and extracorporeal therapy), cefiderocol dosages and plasma concentration-time curves are shown in Figure 1.

### 2.3. Pharmacokinetics

Cefiderocol trough concentrations (C_min_) ranged between levels of 25–70 mg/L, and by this were substantially above the targeted MICs (see Figure 1). Based on these measurements, we simulated the course of the individual cefiderocol concentrations assuming first-order kinetics. Despite the different extent of dose adaptions either empirically or based on TDM results, all patients maintained cefiderocol concentrations 100% *f*T of at least 1 × MIC (e.g., 2 mg/L = total serum concentration of 4 mg/L with a protein binding of 50%).

Whilst on CRRT, C_min_ were 25 mg/L (>4 × MIC) with a dosing regimen of 2–3 × 1 g cefiderocol infused over 3 h. However, there was a significant decrease in serum levels in the patient treated with cytokine adsorber therapy. Even with at a dosage of 3 × 2 g cefiderocol, the addition of the CytoSorb^®^ therapy resulted in C_min_ of 13 mg/L approximately 3 h after its initiation. The Cytosorb^®^ cartridges were exchanged once daily and thus the next dose of cefiderocol resulted in trough levels of 26 mg/L whilst still on CRRT. All patients showed a cefiderocol half-life of at least 8 h.

## 3. Discussion

Here we present real-life data on clinical pharmacokinetics of cefiderocol in five severely ill patients with septic shock due to multi-resistant Gram-negative bacteria. The patients presented with various degrees of AKI and the need for CRRT or Cytosorb^®^ therapy. Due to extensive inter- and intraindividual differences in volume of distribution, potentially altered protein binding as well as elimination rates, prediction of effective plasma levels for antibiotic agents in critically ill patients is difficult [15]. This makes therapeutic drug monitoring an essential tool which is already recommended.

As cefiderocol is a time dependent antibiotic, it is of major importance for the clinical outcome to maintain plasma concentrations above the MIC by the end of the dosing interval (100% *f*T > MIC) [16,17]. Older beta-lactams such as meropenem or ceftazidime were initially designed to target only 40–50% *f*T > MIC and thus makes it difficult to achieve modern PK/PD-targets (100% *f*T > MIC) with the licensed dosing regimens.

Bearing this modern approach in mind, cefiderocol is the first beta-lactam antibiotic whose dosing regimen is designed to attain a PK/PD target of at least 75% *f*T > MIC by using prolonged infusions over 3 h. As described by pharmacokinetic data from Phase 3 studies (CREDIBLE-CR; APEKS-NP) cefiderocol dosing regimens and application mode already result in drug exposure of 100% *f*T > MIC and therefore fulfills modern antibiotic strategies [10]. Recent studies, as well as expert opinions do favor maintaining a target of at least 100% *f*T > MIC or even 100% *f*T > 4 × MIC to optimize microbiological cure as well as clinical outcome [18]. Even if these targets are not (yet) standard, a 100% *f*T > 4 × MIC seems desirable, especially in the case of infections caused by difficult-to-treat multi-drug resistant Gram-negative pathogens [19,20]. Moreover, attainment of a certain drug level to overcome the mutant-resistant concentration is currently examined in order to prevent the development of resistance [17].

Due to the variable MIC of *Pseudomonas aeruginosa*, we chose the theoretical EUCAST cut-off of 2 mg/L as the targeted value [17,21]. Taking into account a cefiderocol protein-binding of 50% a total C_min_ of 4 mg/L (equals a free fraction of approx. 2 mg/L) would be adequate to maintain a PK/PD target of 100% *f*T > MIC. When aiming for higher targets (100% *f*T > 4 × MIC) a total C_min_ of 16 mg/L would be sufficient. None of the initial and empirically adapted dosing regimens resulted in suboptimal drug exposure when aiming for the conservative target of 100% *f*T > 1 × MIC (overall C_min_ > 4 mg/L). Additionally, in concordance with data of healthy volunteers, cefiderocol showed linear pharmacokinetics with prolonged half-lives up to 12 h dependent on renal function [22].

Apparently, in two of the five critically ill patients, initial eGFR values of 60 and 80 mL/min/1.73 m^2^ (patient #2, #3) were not reliable to guide cefiderocol dosing sufficiently. The trough values of these patients were 36.8 and 70 mg/L indicating a prolonged half-life for cefiderocol and an overestimation of their renal function. This is in accordance with previous findings showing highly variable beta-lactam clearances and significant discrepancies of CKD-EPI eGFR and measured GFR in critically ill patients [23,24]. Whilst on CRRT sufficient C_min_ (>13 mg/L) were achieved with a dosing regimen of 2–3 × 1 g cefiderocol infused over 3 h. This corresponds to the approximations based on cefepime data (2 × 1 g during CRRT), assuming similar pharmacokinetics to cefiderocol [6]. Furthermore, the volume of distribution was increased within the patients on ECMO which is represented by the lower peak concentrations in these patients (#3, #5). This might be due to the higher degree of critical illness rather than effected by ECMO itself [25]. As shown for other beta-lactams [26], the addition of a CytoSorb^®^-therapy to CRRT resulted in a substantially higher initial drug elimination with a cefiderocol C_min_ of 13 mg/L approximately 3 h after CytoSorb^®^ initiation. This is of utmost importance considering that source control and appropriate antibiotic therapy remain the cornerstones of successful sepsis therapy today [18,27]. These findings, confirm the high variability in drug elimination in the vulnerable cohort of critically ill patients. Furthermore, as previous studies could show a positive effect of TDM on patient outcome, measurement of cefiderocol exposure gains more importance [28].

The resistant isolate in patient #4 could be linked to a previously isolated sub-clone from a throat swab. This finding was not known to the treating physicians, which is why it was not taken into account when choosing the empirical therapy. Thus, no proven emergence of resistance during therapy with cefiderocol was observed in this study. A limitation of our study is the small number of patients included. To confirm our findings larger studies with well characterized patient populations are needed. These should include patients from different ethnicities [29,30], patients with liver impairment and protein deficiency [31] as well as patients with chronic hemodialysis vs. acute renal failure.

## 4. Materials and Methods

### 4.1. Patients and Setting

We aimed to study the pharmacokinetics of cefiderocol in critically ill patients with AKI or CRRT and with a proven infection by a cefiderocol-susceptible pathogen who were treated at the Department of Intensive Care Medicine at the University Medical Center Hamburg-Eppendorf, Germany. The hospital is a tertiary care center with 140 intensive care beds for adult patients and offers all methods of extracorporeal organ replacement therapies as well as solid organ and bone marrow transplant capabilities. Individual patient treatment was based on national and institutional standards for the management of septic shock at the discretion of the treatment team. Basic demographic data such as weight and sex as well as routine laboratory data (e.g., creatinine level, leucocyte counts, c-reactive protein) were extracted from the patient data management system (ICM, Dräger, Lübeck, Germany).

Additionally, data on clinical performance such as the need for CRRT or concomitant catecholamine therapy were collected. Septic shock was defined according to the 2016 Third International Consensus Definition for Sepsis and Septic Shock [32]. Severity of illness was evaluated by sequential organ failure assessment (SOFA) and simplified acute physiology score (SAPS) [33,34]. Initial dosing of cefiderocol was performed according to the manufacturers’ recommendation for patients with normal or decreased renal function. Subsequently, dose adjustment was guided by TDM. Cefiderocol (1 or 2 g) was reconstituted with 10 mL of normal saline and further diluted. Prolonged infusion was performed over three hours via a central line.

Based on previous experiences and in-vitro data [26] we intensified TDM after the initiation of cytokine adsorbent therapy (CytoSorb^®^, CytoSorbents Corporation, Monmouth Junction, NJ, USA). The data acquisition was reviewed and approved by the Ethics Committee of the Hamburg Chamber of Physicians (#WF003/21) and the study was conducted according to the guidelines of the Declaration of Helsinki.

### 4.2. Cefiderocol Quantification

According to the department’s regulations for TDM, trough levels (C_min_) of cefiderocol were drawn routinely. Samples for TDM were collected on day two of cefiderocol treatment initiation and repeated with occurrence of either dose adjustment or alteration in renal function including initiation of extracorporeal organ support. The specimens were centrifuged (3000× *g*, 15 min) and the supernatant stored at −20 °C until further analysis. Same day cefiderocol quantification was performed by high performance liquid chromatography with UV-detection (Nexera-I 3D plus, Shimadzu; Duisburg, Germany) after protein-precipitation with acetonitrile/methanol (1:1) with subsequent centrifugation of the samples. The supernatant was further diluted with 500 µL of HPLC solvent A (0.1% formic acid in water HPLC grade) and injected onto the HPLC-UV. Chromatographic analysis was performed using a gradient of solvent B (0.1% formic acid in acetonitrile) with a flow-rate of 0.35 mL/min and an autosampler and column temperature of 10 and 45 °C, respectively. Cefiderocol was detected at a wavelength of 300 nm. As previously shown [11], the cefiderocol peak could be clearly distinguished from other cephalosporins by HPLC-UV. The method was validated according to the requirements of the German Society for Toxicology and Forensic Chemistry [16]. With a deviation of ±15% along the concentration range, the accuracy of the samples analyzed on different days was within the acceptance criteria. The calibration curves were linear over the concentration range from 4 to 160 mg/L with a correlation coefficient of 0.999. Of great importance is the fact that stability of patient samples is highly dependent on the storage conditions, with only 2 h when stored at room temperature, one day when stored at −20 °C and up to 31 days when stored at −80 °C [11].

### 4.3. Cefiderocol Susceptibility Testing

Cefiderocol MIC testing was performed using pre-prepared microbroth dilution plates (Sensititre^TM^, Thermo Fisher Scientific; Bremen, Germany) with cation-adjusted Mueller-Hinton broth with TES (CAMHB) according to the manufacturer’s instructions. Interpretation of susceptibility was performed according to EUCAST criteria V11.0, 2020 [12]. Repeated microbiological sampling and susceptibility testing was performed during the course of treatment to monitor for the emergence of resistance.

### 4.4. Cefiderocol Plasma Time Curve Visualization

To visualize and estimate cefiderocol’s pharmacokinetics, software support (Pharkin, Germany) was used to simulate plasma-concentration-time curves. Half-lives and volume of distribution were estimated according to the patient’s underlying renal function or continuous renal replacement therapy.

## 5. Conclusions

Cefiderocol pharmacokinetics is altered in critically ill patients with AKI and CRRT. In the studied patients, drug clearance was reduced and should be taken into account for cefiderocol dosing. In general, target levels for cefiderocol plasma concentrations will be achieved with current dosing recommendations. Under treatment for up to 9 days, difficult to treat MDR Gram-negative bacteria showed no changes in cefiderocol susceptibility whilst on optimal drug exposure. A note of caution is the finding that cytokine adsorber therapy might reduce cefiderocol concentrations to levels not sufficient to treat the targeted pathogen. Therefore, therapeutic drug monitoring is highly recommended to monitor and guide cefiderocol dosing in critically ill patients.

## Figures and Tables

**Figure 1 antibiotics-10-00649-f001:**
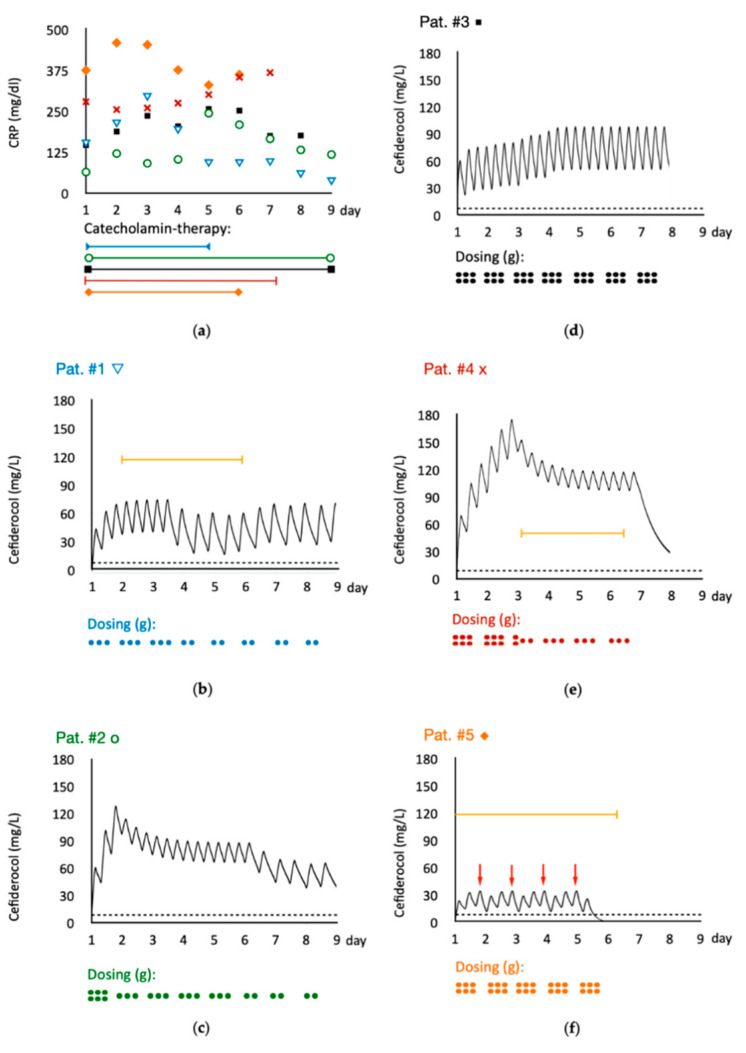
Clinical courses and cefiderocol levels of individual patients. The color scheme is applied consistently for all parts of the figure; blue = patient #1; green = patient #2; black = patient #3; red = patient #4; orange patient #5. (**a**) demonstrates individual courses of C-reactive protein for each patient. In addition, bars at the bottom indicate the duration of catecholamine therapy for each patient. (**b**–**f**) demonstrate the individual cefiderocol levels simulated based on the measured trough levels. Dotted horizontal lines denote the cut-off for sensitivity for the pathogen isolated from each patient according to EUCAST. Yellow bars present the durations of continuous renal replacement therapy. The daily cefiderocol dosage is displayed by dots at the bottom of each patients’ graph. Each dot denotes one gram of cefiderocol. Red arrows in part (**f**) indicate the change of CytoSorb^®^ cartridges with measured consecutive drops of cefiderocol plasma levels.

**Table 1 antibiotics-10-00649-t001:** Demographic data, extracorporeal therapies and microbiological data.

Case	Sex	Age [years]	Weight [kg]	CKD-EPI eGFR (mL/min/1.73 m^2^) or Mode of Renal Replacement Therapy	Cefiderocol Dose [mg/24 h]	Cefiderocol Trough Level [mg/L]	Focus of Infection	SOFA/SAPS	Medical History	Outcome Related to the Event	Pathogen Isolated	MIC [mg/L]	
												First Available Isolate	Last Available Isolate
1	m	35	75	CVVHD DFR 2.0 L/min *	2000	25.2 (day 2)	CAP	11/49	PAH; interstitial lung disease; chronic renal impairment	CC + MC	*P. aeruginosa*	0.5	0.25
				CVVHD DFR 2.0 L/min *	2000	32.0 (day 4)							
2	f	60	60	67	3000	70 (day 1)	HAP	4/33	Esophagectomy, IPA, mediastinitis	MC	*P. aeruginosa*	0.25	n.a.
				22	2000	49 (day 7)							
3	m	76	85	84	6000	36.8 (day 1)	HAP; Primary Sepsis	9/49	COVID-19; ARDS, ECMO	MC	*A. baumannii*	0.25	0.25
				85	6000	43.2 (day 3)							
				81	6000	59.5 (day 4)							
4	m	53	60	28	6000	42 (day 1)	Primary Sepsis	14/68	ASCT; GvHD c/I; invasive mold infection; HHV-6 encephalitis; BKV-cystitis	died	*P. aeruginosa*	0.125	16
				CVVHD DFR 2.4 L/min *	3000	>100 (day 3)							
5	m	55	90	CVVHD DFR 2.4 L/min * + CytoSorb^®^ **	6000	26 (day 1) 13 (day 2)	HAP	19/59	COVID-19; ARDS, ECMO	died	*A. baumannii*	0.25	n.a.

ARDS = acute respiratory distress syndrome; ASCT = allogenic stem cell transplantation; BKV = human papilloma virus; CAP = community acquired pneumonia; CC = clinical cure; COVID-19 = Coronavirus disease 19; CVVHD = continuous venovenous haemodialysis; DFR = dialysate flow rate; eGFR = estimated glomerular filtration rate calculated by CKD-EPI; f = female; GvHD c/l = graft-versus-host disease cutaneous/lung; HAP = hospital acquired pneumonia; HHV-6 = human herpes virus 6; IPA = invasive pulmonary aspergillosis; m= male; MC = microbiological cure; MIC = minimal inhibitory concentration; n.a. = not applicable; SAPS = simplified acute physiology score; SOFA = sequential organ failure assessment; PAH = pulmonary hypertension; * used with the Ultraflux V600S filter with a 1.4 m^2^ surface area (Fresenius Medical Care, Germany); ** installed in series with the CVVHD, with a blood flow of 200 mL/min (CytoSorbents Corporation, Monmouth Junction, NJ, USA).

## Data Availability

Data may be requested on individual request from C. König (ch.koenig@uke.de).

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
