# Peer review of "Cefiderocol in Critically Ill Patients with Multi-Drug Resistant Pathogens: Real-Life Data on Pharmacokinetics and Microbiological Surveillance"

_antibiotics, 2021, doi:10.3390/antibiotics10060649_

Round 1

Reviewer 1 Report

In this study, König et al., describe 5 cases of critically ill patients with severe septic shock, acute renal failure requiring continuous renal replacement therapies and cytokine absorber therapy. To gain knowledge on the pharmacokinetics of cefiderocol in this vulnerable population, therapeutic drug monitoring was performed, alongside microbiological surveillance. Outcomes included cefiderocol through levels, and clinical and microbiological cure. Despite the small cohort, this is a solid study that provides important real life data, and therefore its contribution to professionals in infectious diseases, pharmacology, and intensive care is significant.

My only concern is regarding the cefiderocol resistant isolate isolated from patient #4. What evidence do authors have to state that it “could be linked to a pre-existing sub-clone, thus no real emergence of resistance during therapy with cefiderocol was observed in this study” Lines 254 – 255? Did authors perform whole genome sequencing or PFGE to be able to say those are different strains? How long ago was that sub-clone infecting the patient? More information in this regard is needed.

Author Response

Reply:

We appreciate reviewer #1's comment. The sub-clone in patient #4 had already been isolated in a previous throat swab. This finding was not known to the treating physicians, which is why it was not taken into account in the choice of the (empirical) treatment. The later isolated sub-clone was identical to the initial one. We clarify this in the modified manuscript (line 297-303).

Reviewer 2 Report

Wichmann et al reported the pharmacokinetic study of Cefiderocol for a set of five patient population aiming at deeply investigating the pharmacokinetic profile of the Cefiderocol in ill patients with renal failure and septic shock. The results showed that the pharmacokinetics of the cefiderocol is significantly changed in critically ill patients. Additionally, the suggested manufacturer's dosing of cefiderocol showed to deliver the drug level to above its MIC value. Interestingly, the authors found that using cytokine absorber therapy significantly lower the drug level. Based on these observations, the authors concluded that the therapeutic drug monitoring is highly recommended to be monitored and that the dose of cefiderocol in critically ill patients should be adjusted.

Overall, this is an interesting study which represent deep insights about the cefiderocol pharmacokinetics in the critically ill patients. However, the major concerns in this study is the sample size to draw a conclusion. The authors's conclusion was based on very small sample size with wide variations e.g., in illness (Medical history) and ages. Accordingly, the authors could not determine/estimate the dose required in case of cytokine absorber. As mentioned earlier, the idea behind this study is interesting, however, the study seems to be un-mature and the authors have to apply it for wider population in order to address a more concise conclusion. In addition, the manuscript requires major modifications as detailed below:

1- the abstract is poorly written. The authors should extend it to briefly discuss the changes in pharmacokinetic profile occured and clearly state the results.

2- the introduction, the authors should extend the previously known pharmacokinetics of cefiderocol and their mode of actions e.g Menichetti etal 2020 Clinc. inf Dis, 10.1093/ofid/ofaa439.1498, 10.3390/antibiotics10030242

3- As mentioned before the authors depend on small sample size to draw their conclusion which make it not significant until statistical analysis represented.

4- Additionally, the authors did not perform a full pharmacokinetic study for the patients, e.g,. it would be informative to have PD, %fT>MIC, %fT>MIC,ELF, TOC...

Author Response

Reviewer #2

Wichmann et al reported the pharmacokinetic study of Cefiderocol for a set of five patient population aiming at deeply investigating the pharmacokinetic profile of the Cefiderocol in ill patients with renal failure and septic shock. The results showed that the pharmacokinetics of the cefiderocol is significantly changed in critically ill patients. Additionally, the suggested manufacturer's dosing of cefiderocol showed to deliver the drug level to above its MIC value. Interestingly, the authors found that using cytokine absorber therapy significantly lower the drug level. Based on these observations, the authors concluded that the therapeutic drug monitoring is highly recommended to be monitored and that the dose of cefiderocol in critically ill patients should be adjusted.

Overall, this is an interesting study which represent deep insights about the cefiderocol pharmacokinetics in the critically ill patients. However, the major concerns in this study is the sample size to draw a conclusion. The authors's conclusion was based on very small sample size with wide variations e.g., in illness (Medical history) and ages. Accordingly, the authors could not determine/estimate the dose required in case of cytokine absorber. As mentioned earlier, the idea behind this study is interesting, however, the study seems to be un-mature and the authors have to apply it for wider population in order to address a more concise conclusion. In addition, the manuscript requires major modifications as detailed below:

Reply

We acknowledge the comment of reviewer#2 that the sample size of our study is small. Taken into account that so far less then 1000 patients were included in randomized controlled trials, this case series of severely ill patients presents important information for physicians in the field.

We know that cytokine adsorber therapy is pushed heavily by the industry. Nevertheless, no randomized controlled trials with positive results have been published yet. In general, there is no simple mechanical solution to a complex biological problem and thus we wanted to remind our colleagues to concentrate on the hallmarks of sepsis therapy which is accurate antimicrobial therapy and source control.

1- the abstract is poorly written. The authors should extend it to briefly discuss the changes in pharmacokinetic profile occured and clearly state the results.

Reply

We thank reviewer #2 for her/his advice and have changed the introduction accordingly.

2- the introduction, the authors should extend the previously known pharmacokinetics of cefiderocol and their mode of actions e.g Menichetti etal 2020 Clinc. inf Dis, 10.1093/ofid/ofaa439.1498, 10.3390/antibiotics10030242

Reply

We appreciate the hint of reviewer #2 and have included the valuable citation.

3- As mentioned before the authors depend on small sample size to draw their conclusion which make it not significant until statistical analysis represented.

Reply

Due to the resistance (in Germany) situation, the use of reserve substances such as cefiderocol is rarely necessary, which accounts for the small number of cases. Even worldwide, less than 1.000 patients have been treated with cefiderocol in studies to date. The aim of this study was to describe the PK of cefiderocol in critically ill patients. By nature, these patients show a high variability. As this is a description of a situation and not a comparison of two therapies, statistical analysis is not necessary.

4- Additionally, the authors did not perform a full pharmacokinetic study for the patients, e.g,. it would be informative to have PD, %fT>MIC, %fT>MIC,ELF, TOC...

Reply

As stated in the manuscript, this study was intended to describe the PK (pharmacokinetics) of cefiderocol in the clinical setting of different entities of infection. Therefore, data on PD (pharmacodynamics) was not the aim of the study and also can not be provided in retrospect. Additionally, evaluation of PD in-vivo with clinical microbiological specimens is difficult since blood cultures are usually drawn once per day.

Data on %fT>MIC are included in the original mansucript (line 215-223).

ELF (epithelial lining fluid): The determination of substances in ELF is technically extremely challenging and reliable results can only be expected in specialized working groups. (Collection of samples must be highly standardized; the calibration by serum/ELF-creatinine-quotient). Furthermore, the PK determination in ELF requires sequential measurements which are not tolerable for severely ill patients. Finally, not all patients suffered from pneumonia, which makes the determination of cefiderocol in ELF meaningless.

TOC (Test of cure): We provide patient individual data on microbiological and clinical cure in the results section and in Table #1. However, we would like to state that this study was not intended as study for drug approval and thus clinical cure (EMA-requirement for such studies) or microbiological cure (FDA-requirement) was not part of the study protocol.

Reviewer 3 Report

General comments

As cefiderocol is a novel drug, the clinical data on its effectiveness against MDR Gram-negative bacteria in critically ill patients is lacking. This makes the results of this study (case series) relevant. However, as indicated at the end of the discussion session, the study's major limitation is the small number of patients included. Also, there was great clinical inter-individual variability among patients. Therefore, all the conclusions should be rephrased accordingly, avoiding firm statements and rather using a suggesting tone. This is also important when concluding based on ONE patient that something is 'Of utmost importance".

The exact design of the study should be given in the methods section.

Specific comments

Replace male with man or men, female with woman or women

line 57-60, Here, we report the results of therapeutic drug monitoring for cefiderocol in a cohort of five patients with infections caused by multi-drug resistant Gram-negative pathogens with severe septic shock, acute renal failure (AKI) requiring CRRT and cytokine absorber therapy. - it is confusing as from this sentence and from the individual presentation of all patients in section Results, one may conclude that all five patients required CRRT  which was not the case according to the sentence in line 70-71,

similar counts for Cytokine absorber therapy ...please be more precise

line 70-71, Continuous renal replacement therapy (CRRT) performed as haemodialysis was applied in three - according to the CRRT is and additional CytoSorb® therapy in one patient. 

line 66, four were male replace with  four were men

line 72/114, male and female replace with man and woman

line 207, very difficult replace with difficult

line 221,  erase the dot after pathogens

line 242, [6].Furthermore,   replace with  [6]. Furthermore,

line 242, within the patients, replace with in the patients

line 255, real emergence replace with proven emergence

line 258, [27,28] ,patients replace with [27,28], patients

line 274, for. CRRT - erase the dot after for

line 323, was estimated replace with were estimated

Author Response

Reviewer #3

As cefiderocol is a novel drug, the clinical data on its effectiveness against MDR Gram-negative bacteria in critically ill patients is lacking. This makes the results of this study (case series) relevant. However, as indicated at the end of the discussion session, the study's major limitation is the small number of patients included. Also, there was great clinical inter-individual variability among patients. Therefore, all the conclusions should be rephrased accordingly, avoiding firm statements and rather using a suggesting tone. This is also important when concluding based on ONE patient that something is 'Of utmost importance".

The exact design of the study should be given in the methods section.

Reply

We appreciate the comments of reviewer #3 and have changed the manuscript accordingly.

For example “Of utmost importance” has been changed to “A note of caution”.

With regard to the special comment, we would like to thank reviewer #3 for her/his careful reading and correction of our manuscript.

Specific comments

Replace male with man or men, female with woman or women

line 57-60, Here, we report the results of therapeutic drug monitoring for cefiderocol in a cohort of five patients with infections caused by multi-drug resistant Gram-negative pathogens with severe septic shock, acute renal failure (AKI) requiring CRRT and cytokine absorber therapy. - it is confusing as from this sentence and from the individual presentation of all patients in section Results, one may conclude that all five patients required CRRT  which was not the case according to the sentence in line 70-71,

similar counts for Cytokine absorber therapy ...please be more precise

line 70-71, Continuous renal replacement therapy (CRRT) performed as haemodialysis was applied in three - according to the CRRT is and additional CytoSorb® therapy in one patient. 

Reply

We acknowledge that this part has to be clarified. To do so, we have added the numbers of individual therapies for each therapy:

Line 65-68: …of five patients with infections caused by multi-drug resistant Gram-negative pathogens with severe septic shock, acute renal failure (AKI) requiring CRRT (n=3) and additional cytokine absorber therapy (n=1).

line 66, four were male replace with  four were men

has been changed

line 72/114, male and female replace with man and woman

has been changed

line 207, very difficult replace with difficult

has been changed

line 221,  erase the dot after pathogens

has been changed

line 242, [6].Furthermore,   replace with  [6]. Furthermore,

has been changed

line 242, within the patients, replace with in the patients

has been changed

line 255, real emergence replace with proven emergence

has been changed

line 258, [27,28] ,patients replace with [27,28], patients

has been changed

line 274, for. CRRT - erase the dot after for

has been changed

line 323, was estimated replace with were estimated

has been changed

Round 2

Reviewer 2 Report

Thanks to the authors for their reply. The manuscript has been modified and improved. However, there are some points still need to be modified:

- the abstract still need to be modified. The authors should extend it to briefly discuss the changes in pharmacokinetic profile occured and clearly state the results.

- Although the authors have mentioned in their reply that they have modified the into part and have included the suggested citation, the modified version does not include such modifications. Again, the authors did not modify the introduction part as suggested. The authors should extend the previously known pharmacokinetics of cefiderocol and their mode of actions e.g Menichetti etal 2020 Clinc. inf Dis, 10.1093/ofid/ofaa439.1498, 10.3390/antibiotics10030242

Author Response

Reply to reviewer #2:
1) the abstract still need to be modified. The authors should extend it to briefly discuss the changes in pharmacokinetic profile occured and clearly state the results.

Reply 1):
We thank the reviewer for the helpful comment and have modified the abstract accordingly (line 20-23 ): “As expected for a cephalosporin with predominantly renal elimination the maintenance dose could be reduced in patients with renal impairment or on continuous renal replacement therapy. The manufacturer's dosing instructions were sufficient to achieve a drug level well above the MIC.”

2) Although the authors have mentioned in their reply that they have modified the into part and have included the suggested citation, the modified version does not include such modifications. Again, the authors did not modify the introduction part as suggested. The authors should extend the previously known pharmacokinetics of cefiderocol and their mode of actions e.g Menichetti etal 2020 Clinc. inf Dis, 10.1093/ofid/ofaa439.1498, 10.3390/antibiotics10030242

Reply 2):
We would like to apologize for not recognizing that “Menichetti etal 2020 Clinc. inf Dis, 10.1093/ofid/ofaa439.1498, 10.3390/antibiotics10030242” consisted of three references.

Regarding the reference Katsube et al (doi: 10.1093/ofid/ofaa439.1498): we have included the information into the introduction (Line 58-60) and also discuss it in the context of our findings later (line 222-225).

Regarding the reference Menichetti et al 2020 Clinc. Inf Dis, (First author Marco Falcone): As stated in our last point-to-point reply we have included it into the references (Ref. #9). The reviewer is right that this publication gives very important information on the real life use of cefiderocol in critically ill patients.

Regarding the reference Zimmer et al (doi: 10.3390/antibiotics10030242). This is a publication from our own group, which is already included and discussed in the manuscript (Ref. #11).